# Anti-Cancer Potential of Linear β-(1→6)-D-Glucan from *Agaricus bisporus* on Estrogen Receptor-Positive (ER+) Breast Cancer Cells

**DOI:** 10.3390/molecules29194781

**Published:** 2024-10-09

**Authors:** Renata Rutckeviski, Claudia Rita Corso, Aline Simoneti Fonseca, Mariane Londero Rodrigues, Yony Román-Ochoa, Thales Ricardo Cipriani, Luciane Regina Cavalli, Silvia Maria Suter Correia Cadena, Fhernanda Ribeiro Smiderle

**Affiliations:** 1Faculdades Pequeno Príncipe, Curitiba 80230-020, PR, Brazil; renatarut@hotmail.com (R.R.); claudia_rcorso@hotmail.com (C.R.C.); 2Instituto de Pesquisa Pelé Pequeno Príncipe, Avenida Munhoz da Rocha, 490, Curitiba 80035-000, PR, Brazil; simonetialine@gmail.com; 3Departamento de Bioquímica Biologia Molecular, Universidade Federal do Paraná, Curitiba 81531-980, PR, Brazil; marianerodrigues@ufpr.br (M.L.R.); silvia.cadena@ufpr.br (S.M.S.C.C.); 4Department of Oncology, Lombardi Comprenhensive Cancer Center, Georgetown University, Washington, DC 20007, USA; luciane.cavalli@pelepequenoprincipe.org.br

**Keywords:** *Agaricus bisporus*, β-(1→6)-D-glucan, MCF-7, breast cancer cell

## Abstract

Mushroom β-D-glucans can be isolated from several species, including the widely consumed *Agaricus bisporus.* Besides immunomodulatory responses, some β-D-glucans may exhibit direct antitumoral effects. It was previously observed that a β-(1→6)-D-glucan (BDG16) has indirect cytotoxicity on triple-negative breast cancer cells. In this study, the cytotoxicity of this same glucan was observed on estrogen receptor-positive (ER+) breast cancer cells (MCF-7). Cell viability was determined by multiple methods to assess metabolic activity, lysosomal membrane integrity, and adhesion capacity. Assays to evaluate cell respiration, cell cycle, apoptosis, necroptosis, and oxidative stress were performed to determine the action of BDG16 on MCF-7 cells. A gradual and significant cell viability reduction was observed when the cells were treated with BDG16 (10–1000 µg/mL). This result could be associated with the inhibition of the basal state respiration after incubation with the β-D-glucan. The cells showed a significant arrest in G1 phase population at 1000 µg/mL, with no induction of apoptosis. However, an increase in necrosis and necroptosis at the same concentration was observed. No difference in oxidative stress-related molecules was observed. Altogether, our findings demonstrate that BDG16 directly induces toxicity in MCF-7 cells, primarily by impairing mitochondrial respiration and promoting necroptosis. The specific mechanisms that mediate this action are being investigated.

## 1. Introduction

The potential of mushroom-derived β-glucans in the prevention and treatment of various types of cancer is widely recognized and utilized in Eastern populations [1]. Over the last few decades, extensive research has been conducted, demonstrating the ability of such polysaccharides to modulate immune system cells, eradicate tumor cells and/or to act synergistically with anti-cancer drugs [2,3]. The β-D-glucans are molecules that differ in their primary structure, molecular weight, branching frequency, and solubility, which can lead to variations in their physiological effects [4]. An association between the chemical structure and the activities of polysaccharides has been established; however, definite conclusions about which mushroom β-glucan exhibit the highest bioactivity remain elusive.

*Agaricus bisporus* mushrooms are the most widely consumed mushrooms worldwide, and their nutritional composition can enhance the immune cell system and be used as a supplement for the treatment of several types of diseases [5]. The linear β-(1→6)-D-glucan isolated from *A. bisporus* was found to stimulate the expression of pro-inflammatory genes in THP-1 macrophages, a human monocytic cell line differentiated to macrophages [6], and to increase the sensitivity of triple-negative breast cancer cells to doxorubicin [3]. A branched β-(1→6)-D-glucan isolated from *Armillariella mellea*, containing O-3 substitutions, presented the ability to inhibit the viability of colon cancer cells in vitro and in vivo by converting the tumor-associated macrophages into an M1-like phenotype [7].

Since 1985, a β-glucan known as Lentinan^®^ has been licensed as a drug for treating gastric and lung cancer in Japan and China. Recently, the use of mycotherapy as a complementary treatment to conventional medicine has been increasingly investigated as a novel therapeutic approach [8]. However, few studies have been conducted addressing specifically the use of β-glucans in the management of breast cancer. Therefore, experimental investigations are critical to determine the application of such compounds in breast cancer treatment protocols.

Breast cancer is the most prevalent cancer among women worldwide [9]. Although significant advances have been achieved in the treatment modalities for these tumors, including the use of target therapies, a high mortality rate is still observed [10]. Chemotherapy remains the most used therapy in several subtypes of breast cancer, isolated or in combination with target therapies. The use of different chemotherapeutic agents and regimens has been effective for the initial response; however, tumor resistance to these drugs eventually occurs. In addition, there is a high occurrence of treatment side effects, which can be prolonged and significantly affect patients’ quality of life [11,12].

In this context, several efforts have been put towards the search for therapeutic agents that could specifically target tumor cells, increase the sensitivity of tumor cells to chemotherapy, and reduce the occurrence of treatment side effects. One of the most promising complementary approaches observed is the use of extracts and molecules obtained from mushrooms [8,13]. Among the molecules that can be isolated from edible mushrooms, the β-glucans are the ones with the most therapeutic potential, considering their different structural features and diverse biological properties, which can impact distinct and critical signaling pathways in breast cancer [14]. Previously, our group demonstrated an indirect antitumor property of a β-(1→6)-D-glucan on the triple-negative breast cancer cells (MDA-MB-231) through its immunostimulatory activity on macrophages. However, there are no conclusive data on the effects of this linear β-(1→6)-D-glucan on estrogen positive-receptor (ER+) breast cancer cells. Therefore, the main aim of this study is to determine the potential of β-(1→6)-D-glucan on ER+ cells, using the MCF-7 cell line as a model, to understand the direct and indirect effects of such polysaccharide on ER+ antitumor therapy.

## 2. Results and Discussion

The polysaccharide used in this study was a linear β-D-glucan with (1→6)-linkages that was obtained from the mushroom *A. bisporus* through aqueous extraction [15]. As described by the authors and tested in the present study, this β-D-glucan is soluble in water and presents M_w_ 8.26 × 10^4^ g/mol (dn/dc = 0.197 mL/g). In this study, the tested β-(1→6)-D-glucan was named BDG16.

### 2.1. BDG16 Decreases the Metabolic Activity of MCF-7 Cells

To assess the BDG16 toxicity on the MCF-7 breast cancer cells, the first step was to determine cell viability using three different approaches: evaluation of metabolic activity, lysosome membrane integrity, and adhesion capacity. The cell viability evaluated by the MTT assay showed a reduction of approximately 31% after treatment with 10 µg/mL of BDG16 for 48 h and reduced viability below 50% with 300 and 1000 µg/mL (Figure 1A). A longer period of treatment (120 h) under the same concentrations did not cause a higher reduction of viability (Figure 1B) when compared to cells treated for 48 h: a reduction of 35% and 47% was observed for MCF-7 cells when they were treated with 300 and 1000 µg/mL (120 h), respectively. This result raises questions about the potential for cell recovery following extended incubation with BDG16.

The cytotoxicity of BDG16 to non-tumorigenic mammary epithelial cells (MCF-10A) was previously evaluated by Rutckeviski et al., 2022, and it was observed that BDG16 was not toxic at concentrations up to 100 µg/mL [3].

Surprisingly, when the cells were evaluated for their lysosome membrane integrity and adhesion capacity, no reduction was observed for both assays at 48 h and 120 h. A neutral red assay revealed that BDG16 caused no harm to lysosomal membranes at all concentrations evaluated (Figure 1C,D); a slight reduction (6% and 10%, respectively) was observed only at 1000 µg/mL, for both incubation periods, on the crystal violet assay (Figure 1E,F).

### 2.2. Cell Respiration

Based on the results of the MTT assay (Figure 1A,B), it can be inferred that BDG16 affects the metabolic functions, as indicated by the decrease in the viability of MCF-7. Therefore, we conducted cellular respiration assays, considering that oxidative phosphorylation is the primary source of ATP in the cells [16]. As shown in Figure 2, the oxygen uptake in the basal state (in the absence of inhibitors or uncouplers) was inhibited (15%) by the polysaccharide (200 µg/mL, 48 h). This inhibition, although not significant, was reflected in the uncoupled state of respiration, characterized by the maximal rate of electron transport in the respiratory chain. The respiration in the leak state (in the presence of oligomycin), resulting from the re-entry of protons into the mitochondrial matrix, was not affected by BDG16 (Figure 2). Considering that the respiration rate during the leak state depends on inner mitochondrial membrane integrity, the absence of the BDG16 effect is compatible with the results of viability assays with neutral red. In these assays, the integrity of lysosomal membranes was evaluated, and it was shown that they were also not affected by BDG16 (Figure 1C,D). These results showed that the inhibition of the basal state by BDG16 seems to be essential for its toxicity in MCF-7 cells. However, further analyses of this cell model are required to determine its effect on the activity of respiratory chain complexes.

### 2.3. Cell-Cycle Evaluation

The crystal violet assay demonstrates cell adhesion and consequently can provide information on cell proliferation. Considering that the use of the highest concentration of BDG16 promoted only a slight reduction in cell proliferation, it can be questioned whether this compound affects cell growth. To address this question, cell-cycle analysis was performed by flow cytometry after treating the cells with BDG16 (10, 30, 100, 300, and 1000 µg/mL) for 48 h.

As shown in Figure 3A, BDG16 showed a tendency to increase the population of MCF-7 cells in the G1 phase after 48 h. As observed in the crystal violet assay, only at 1000 µg/mL, the cells showed a significant arrest in the G1 phase (14%) compared to the control.

### 2.4. Apoptosis and Necroptosis Evaluation

A decrease in cell viability by 8% and 10% was observed in the MCF-7 cells treated with BDG16 at 300 and 1000 µg/mL for 48 h, respectively (Figure 3B). A significant induction of necrosis was observed when cells were treated with 1000 µg/mL of BDG16. No induction of apoptosis was observed in the breast cancer cells after 48 h of treatment with BDG16 (Figure 3B). In addition, the treatment with BDG16 (100, 300, 500, and 1000 μg/mL) did not alter the expression of the *BCL-2* gene, which codes an anti-apoptotic regulatory protein (Appendix A) [17]. These results are compatible with the already known unharmed cell effect of mushroom polysaccharides, even to cancer cells, which are usually more resistant and develop apoptosis evasion mechanisms [18]. On the other hand, the results of this study indicate cell death after exposure to BDG16, demonstrated by the increase of necrosis at the highest concentration (Figure 3B). 

Recent studies demonstrated that the necroptosis pathway may be another mechanism of cell death when apoptosis is compromised [19]. Necroptosis is a form of regulated necrosis, mediated by RIPK1 (receptor-interacting protein kinase 1), RIPK3, and MLKL (mixed lineage kinase domain-like pseudokinase), and it is repressed by Necrostatin-1 (NEC-1), due to its ability to inhibit RIPK1 activity [19]. Based on this mechanism, further investigation was conducted by incubating the MCF-7 cells treated with BDG16 for 48 h in the presence and absence of NEC-1. The results showed a significant reversed effect on the cell viability at 1000 µg/mL (Figure 3C), indicating that protein kinase 1 (RIPK1) might be involved in the cytotoxicity mediated by BDG16.

### 2.5. Oxidative Stress Analysis

Based on the MTT and cell respiration assays (Figure 1 and Figure 2), it seems that BDG16 affects mitochondrial activity. The inhibition of the basal state may result from a decrease in the rate of electron transport in the respiratory chain, which in turn may increase ROS production [20]. Therefore, additional analyses were performed to investigate the oxidative status of the MCF-7 cells incubated with BDG16. ROS levels did not change after incubation with the polysaccharide (Figure 4A). Interestingly, it was previously demonstrated that BDG16 can reduce superoxide dismutase (SOD) activity in MCF-7 cells (at 10 and 1000 µg/mL), which is the first line of defense against oxidative stress, transforming superoxide anion in hydrogen peroxide [3]. On the other hand, an increase in the total antioxidant activity (T-AOC) was observed when the cells were incubated with BDG16 at 30, 300, and 1000 µg/mL (19%, 50%, and 42%, respectively) (Figure 4B). These results suggested that BDG16 possesses antioxidant activity, which could prevent the ROS increase resulting from respiration inhibition. To investigate this further, we evaluated the activity of the antioxidant enzymes GST and GPx. GST presents an essential role in the detoxification of ROS along with reduced glutathione (GSH) and is also an important enzyme in the phase II metabolism of xenobiotics, especially chemotherapeutics [21]. Although we observed an increase in the GST activity following incubation with BDG16 (300 µg/mL), this effect was not statistically significant (Figure 4C). In addition, no difference was observed in GPx activity on the cells (Figure 4D). GPx has an important action in the elimination of hydroperoxides (lipid peroxidation products), transforming the reduced form of glutathione (GSH) into the oxidized form (GSSG) [21].

Altogether, the results observed in this study demonstrate that BDG16 is cytotoxic to MCF-7 cancer cells, and this effect is likely related to necroptosis mechanisms. Extensive research has been conducted on the role of mitochondria in necroptosis, but it remains unclear whether inhibiting respiration directly triggers such a mechanism of cell death [22,23,24,25]. Basit et al. (2017) [22] investigated this in BRAF^V600E^ human melanoma cells and showed that the inhibition of respiratory chain complex I led to a series of events, including mitochondrial permeability transition and depolarization of the mitochondrial membrane potential, resulting in combined necroptotic/ferroptotic cell death. However, their study showed an increase in ROS levels, which was not observed in the present work. This suggests that the mechanism by which BDG16 induces necroptosis is complex and requires further investigation to elucidate the mitochondria contribution. According to Dhuriya and Sharma (2018) [26], necroptosis can be initiated by TNF superfamily receptors, and extrinsic necroptosis is stimulated by TNF-α, culminating with the inactivation of caspase-8 and activation of RIPK1 and RIPK3. This event promotes the phosphorilation of pseudokinase MLKL (mixed lineage kinase domain-like protein), which plays a key role in the induction of necroptosis [26]. It is known that mushroom β-glucans can stimulate the production of pro-inflammatory cytokines, such as TNF-α, and it was demonstrated that BDG16 increased the secretion of such cytokines in THP-1 macrophages [3]. Furthermore, the indirect antitumor effect of such molecules was demonstrated by many researchers due to stimulation of the immune system. However, some authors have observed direct cytotoxic effects of β-glucans on cancer cells or proliferation inhibition [27]. Such mechanisms of action include (1) inhibition of phosphorylation of several signaling molecules as demonstrated in lung cancer cells [28], (2) modulation of expression of some cancer-related genes in both canine and human tumor cells [29], and (3) inhibition of proliferation of breast cancer cells when β-glucan was used in hypoxic conditions [27]. Although up to this date, necroptosis has not been directly associated with the treatment of cancer cells with β-glucans, further and more comprehensive investigation should be performed to expand the use of such compounds in cancer therapy.

A clinical study was developed with breast cancer patients using a soluble β-glucan named SBG [30], and the researchers evaluated the “unfavorable side effects” and the “beneficial treatment effects when SBG was given in combination with standard antibody and chemotherapy to patients”. This clinical trial was completed in 2010 and the outcomes are not provided. Many other clinical trials have been performed using β-glucans to concomitantly treat cancer patients based on the evidence that such compounds are stimulators of immune cells and, therefore, help the immune system to kill tumor cells. However, the studies do not evaluate the direct effect of β-glucans on tumors. This study demonstrates that a soluble and linear (1→6)-β-D-glucan (BDG16) was able to arrest MCF-7 cells in the G1 phase and stimulate death by necroptosis. It has been confirmed that BDG16 did not cause oxidative imbalance, as it acts as an antioxidant compound. Its mechanism of action is still being investigated to provide more information and support clinical trials.

## 3. Material and Methods

### 3.1. Reagents

Dulbecco’s modified eagle medium (DMEM-F12) and fetal bovine serum (FBS) were purchased from Gibco^TM^ (Paisley, United Kingdom) Dimethylsulfoxide (DMSO), Triton; necrostatin-1 was purchased from Sigma-Aldrich (São Paulo, SP, Brazil). Penicillin/streptomycin, trypsin, Annexin V-FITC, and 7-AAD were purchased from Invitrogen-Thermo Fisher (Waltham, MA, US, EUA). MTT and RNase were acquired from Ludwig Biotecnologia (Alvorada, RS, Brazil). Paraformaldehyde and crystal violet were purchased from Vetec (Duque de Caxias, RJ, Brazil). Acetic acid and ethanol were purchased from Synth (Diadema, SP, Brazil). All other reagents used were of the highest commercially available purity.

### 3.2. Fungal Material and Preparation of β-Glucan

β-(1→6)-D-glucan (named in this study as BDG16) obtained from *A. bisporus*, was extracted and chemically characterized in accordance with the protocol previously described by Smiderle et al. (2013) [6], and the second batch produced by Román et al. (2017) [15] was used in this study. Briefly, *A. bisporus* fruiting bodies (120 g) were first extracted with cold water (2 L). The extract was separated, and the insoluble residue was extracted with water (1.5 L) at 100 °C for 4 h (4×). The hot aqueous extract was then concentrated under reduced pressure, and the polysaccharides were precipitated by the addition of 3 volumes of cold ethanol (3:1; *v*/*v*), followed by centrifugation at 8000 rpm at 5 °C for 20 min. Then, small particles were removed by dialysis using tap water for 24 h (6–8 kDa cut-off^®^ membrane; Spectra/Por, Regenerate Cellulose Membrane, Darmstadt, Germany). The crude extract was subjected to freezing and slowly thawing several times until total precipitation of cold-water insoluble polysaccharides. Such polysaccharides were separated from the soluble fraction by centrifugation (8000 rpm at 5 °C, for 20 min), and later they were treated with dimethylsulfoxide (at 50 mg of sample/ mL of solvent) at 60 °C for 2 h, dialyzed using tap water for 48 h (6–8 kDa cut-off membrane), and lyophilized to give rise to BDG16 [15].

BDG16 (5 mg) was weighed, solubilized in 0.5 mL of phosphate-buffered saline (PBS), and heated while stirring until complete solubilization. Dilutions of this initial solution were performed using PBS (10, 30, 100, 300, and 1000 µg/mL). 

### 3.3. Estrogen Positive-Receptor (ER+) Breast Cancer Cell Line

ER+ cells (MCF-7) were obtained from the Rio de Janeiro Cell Bank (Rio de Janeiro, Brazil). MCF-7s maintain several characteristics, like mammary epithelium. This cell line has an epithelial-like morphology, and monolayers form dome structures due to fluid accumulation between the culture dish and cell monolayer.

### 3.4. Metabolic Activity Evaluation by MTT Assay

The BDG16 cytotoxicity was evaluated in the MCF-7 breast cancer cell line by the MTT test, according to Mosmann (1983) [31], with minor changes. Initially, 1 × 10^4^ cells/well were let to adhere for 24 h into 96-well plates and incubated with BDG16 at 10, 30, 100, 300, and 1000 µg/mL and dissolved in PBS for 48 h. The procedure was repeated in another plate; however, after 72 h of treatment, the medium was replaced with the same concentrations of BDG16, and the plate was incubated for a further 48 h, totalizing 120 h of treatment.

Three hours before the conclusion of the incubation period, MTT (5 mg/mL in PBS) was added to each well, and the cells were incubated for 3 h at 37 °C. Afterward, the MTT solution was carefully removed and DMSO was added to solubilize purple formazan crystals. The absorbance was read at 595 nm in a microplate reader (Spectrophotometer Biotek^®^, Wellesley, MA, USA, EUA). Analyses were conducted in three independent experiments in quadruplicated wells. Cell viability (metabolic activity) was expressed as a percentage of control cells (vehicle group).

### 3.5. Lysosome Membrane Integrity Evaluation by Neutral Red Assay

MCF-7 cells (1 × 10^4^ cells/well) were seeded and treated as described (Section 3.4) for 48 h and 120 h. After this period, the medium was aspirated, and adhered cells were washed with pre-warmed PBS. Neutral red (NR) (100 μL) dissolved in culture medium (0.04 mg/mL) was added to each well, and the plates were incubated for 2 h at 37 °C. The NR medium was removed, and a solution of 50% ethanol and 1% glacial acetic acid was added to each well. Plates were shaken for 10 min before absorbance measurement at 540 nm (Spectrophotometer Biotek^®^, Wellesley, MA, USA, EUA). Analyses were conducted in two independent experiments in quadruplicated wells. Cell viability (measured by lysosome membrane integrity) was expressed as a percentage of control cells (vehicle group). 

### 3.6. Cell Adhesion Evaluation by Crystal Violet Assay

The same plates used for the neutral red assay were used to measure the adhesion capacity of the cells. After measuring cell viability (Section 3.5), the cells were washed twice with pre-warmed PBS and fixed with paraformaldehyde 2% for 30 min. Then, the paraformaldehyde was removed, and the remaining cells were stained for 10 min with a crystal violet (CV) solution (0.25 mg/mL). After removal of the CV solution, the plates were washed twice with 150 µL of ultrapure water. Then, 100 µL of acetic acid (33%) was added, and the plate was incubated for 30 min on a plate shaker (Corning^®^ LSE™, Sigma-Aldrich, São Paulo, SP, Brazil). The absorbance was read at 570 nm in a microplate reader (Spectrophotometer Biotek^®^, Wellesley, MA, USA, EUA). Analyses were conducted in three independent experiments in quadruplicated wells. Cell viability (measured by the adhesion capacity) was expressed as a percentage of control cells (vehicle group).

### 3.7. Measurement of Cell Respiration

For these assays, MCF-7 cells (1 × 10^6^ cells) were seeded in 6-well culture dishes and allowed to adhere for 24 h. Afterward, the medium was replaced with a new one containing BDG16 (200 μg/mL), and the cells were cultured for 48 h. Then, the medium was discarded, and the cells were washed once with PBS before being collected and re-suspended in the medium. Cells were counted, and the viability was determined by the trypan blue exclusion method [32]. Viable cells (1.5 × 10^6^/ mL) were transferred to the chambers of high-resolution oxygraph-2k (OROBOROS Instruments, Innsbruck, Austria), and the oxygen flux was determined at 37 °C under gentle agitation. The cellular respiration was evaluated in the absence of inhibitors and uncouplers (basal state) after the addition of ATP synthase inhibitor oligomycin (2 μg/mL—leak state). Finally, uncoupler FCCP (3.5 μM) was added to achieve maximal respiration (uncoupled state) [33]. The oxygen flow in these states was corrected by subtracting non-mitochondrial respiration, obtained after adding rotenone (2 µM) and antimycin (3 µg/mL), and expressed per million cells [pmol/(s× 10^6^ cells)].

### 3.8. Cell Cycle Analysis

MCF-7s (3 × 10^5^ cells/well) were plated in 6-well culture plates and cultured for 24 h. After medium replacement, cells were treated with BDG16 at 10, 30, 100, 300, and 1000 µg/mL for 48 h. For this procedure, cells were detached with trypsin, washed once with PBS, and fixed with 70% ethanol overnight at 4 °C. Then, the cells were collected by centrifugation (300 g, for 7 min), re-suspended in PBS containing 7-AAD (7-Aminoactinomycin D) plus 100 μg/mL RNAse, and incubated for 15 min protected from light. The data was acquired on a BD FACS Canto™ II (Franklin Lakes, NJ, USA), using FlowJo X Software 10.8.1 (BD, Becton, Dickinson and Company, Franklin Lakes, NJ, USA), and conducted in three independent experiments in triplicate wells. Cell viability was expressed as a percentage of control cells (vehicle group).

### 3.9. Evaluation of Apoptosis

MCF-7 cells (3 × 10^5^ cells/well) were plated in 6-well culture plates and cultured for 24 h. After medium replacement, cells were treated with BDG16 at 10, 30, 100, 300, and 1000 µg/mL for 48 h. The treated cells were harvested and washed with PBS. After centrifugation (300 g, for 7 min), the cell pellets were re-suspended in binding buffer and stained with Annexin V-FITC and 7-AAD for 15 min, protected from light. The data was acquired on a BD FACS Canto™ II (Franklin Lakes, NJ, USA) and analyzed using Flowing Software 2.5.1 (Turku Bioscience, Turku, Finland). Analyses were conducted in three independent experiments in triplicate wells. Cell viability was expressed as a percentage of control cells (vehicle group).

### 3.10. Necroptosis

To evaluate necroptosis, MCF-7 cells were plated at a density of 1 × 10^4^ cells/well in a 96-well plate. Cells were incubated with BDG16 at concentrations of 10, 30, 100, 300, and 1000 μg/mL for 48 h in the presence or absence of 50 μM necrostatin-1 (Sigma-Aldrich^®^, St. Louis, MO, USA). Cell viability was evaluated according to Section 3.4 and expressed as a percentage of control cells (vehicle group). Analyses were conducted in three independent experiments in triplicate wells.

### 3.11. RT-qPCR Analysis

The expression of a gene related to apoptosis was assessed in the MCF-7 cell line treated with BDG16 (100, 300, 500, and 1000 μg/mL) for 48 h. RNA was isolated using TriZol reagent and quantified using NanoDrop One (Thermo Fisher Scientific). The complementary DNA (cDNA) synthesis was performed from 1.0 μg of extracted RNA using the High-Capacity cDNA Reverse Transcription kit according to the manufacturer’s instructions (Thermo Fisher Scientific). The cDNAs were diluted at 1:5 and stored at −80 °C until analysis. Housekeeping β-*Actin (ACTB)* gene was used to normalize RNA inputs. The levels of expressed genes were measured by RT-qPCR using the 2^−ΔΔCt^ method [34].

### 3.12. Evaluation of Oxidative Stress Parameters

For the evaluation of oxidative stress parameters, MCF-7 cells (5 × 10^5^ cells/well) were incubated in 6-well culture plates with BDG16 (30, 300, and 1000 μg/mL) for 48 h. After medium removal, the cells were washed with PBS, trypsinized, centrifuged (5 min, 1000 g, 4 °C), resuspended in PBS, and immediately frozen at −80 °C to promote lysis. The lysed cells were used for the measurement of ROS, GPx and GST activities, and T-AOC, as described below.

#### 3.12.1. Total Reactive Oxygen Species (ROS) 

To detect these molecules, 200 μL of lysed cells, prepared as described in Section 3.12, and 150 μL of 0.10 mM DCFH-DA (2′,7′-dichlorodihydrofluorescein diacetate) were incubated for 1 h, protected from light. The presence of ROS in the samples hydrolyzes the reagent DCFH-DA into the final product DCF, which is detected by fluorescence [35].

#### 3.12.2. Determination of Glutathione Peroxidase (GPx) Activity

The method is based on the measurement of the decrease in absorbance at 340 nm, which occurs during the reduction of GSSG (oxidized glutathione) catalyzed by GR (glutathione reductase) in the presence of NADPH. The speed of NADPH oxidation is proportional to the rate of GSSG production from GSH in the presence of peroxide catalyzed by GPx. In a 96-well plate, 10 µL of lysed cells (Section 3.12) was mixed with 130 µL of reaction solution containing 1 mM sodium azide, 1 mM GSH, 0.2 mM NADPH, and 1U of GR. The kinetic reaction was started with 60 µL of 0.25 mM H_2_O_2_ [36]. Absorbance was read at 340 nm for 5 min, and GPx activity was calculated from the slope of these lines as µmols NADPH oxidized per minute. The results were expressed as U/mL, which represents µmol/min/mL of the substrate [36].

#### 3.12.3. Determination of Glutathione S-Transferase (GST) Activity

GST enzymes are responsible for initiating the detoxification of alkylating agents through the -SH group of glutathione. The method for detecting GST activity is based on forming a conjugate between CDNB (1-chloro-2,4-dinitrobenzene) and GST, which is detected by kinetics at a wavelength of 340 nm. For the reaction, 100 µL of lysed cells (Section 3.12) were plated with 200 µL of reaction solution containing 1 mM of CDNB and 5 mM of GSH in 10 mM potassium phosphate buffer [37].

#### 3.12.4. Determination of Total Antioxidant Activity (T-AOC)

The method determines the ferric reducing capacity of the sample by reducing the complex between Fe^+3^ and the TPTZ compound (2,4,6-Tris(2-pyridyl)-1,3,5-triazine) in Fe^+2^, an acidic medium. In a 96-well plate, 25 µL of lysed cells (Section 3.12) were incubated with 265 µL of reaction solution containing 10 mM of TPTZ and 20 mM of FeCl_3_ in 300 mM of acetate buffer (pH 3.6) at 37 °C for 30 min. This reaction produces an intense blue color, which is detected at a wavelength of 593 nm.

### 3.13. Statistical Analysis

The data are presented as the mean ± SD and were analyzed using one-way analysis of variance (ANOVA) followed by Bonferroni’s or Tukey’s post hoc test with GraphPad Prism 6.0 software (GraphPad Software, San Diego, CA, USA). *p* ≤ 0.05 was considered statistically significant.

## 4. Conclusions

Less than 30 clinical studies have been performed in the USA with cancer patients being treated with β-glucans as adjuvant. The clinical trials are based on the immunostimulatory effects of such compounds, and up to this date, little information is available to understand the mechanism of action of β-glucans in cancer treatment. In summary, the results observed in this study demonstrate that a linear β-(1→6)-D-glucan (BDG16) isolated from the mushroom *A. bisporus* alters the metabolic activity of MCF-7 cells, significantly reducing their cell viability. At higher concentrations (1000 µg/mL), BDG16 was able to decrease cell proliferation and cell adhesion by inducing cell cycle arrest and cell death, demonstrated by increased necrosis and necroptosis. Based on our findings, it is evident that fungal β-glucans can exhibit a direct antiproliferative effect against the estrogen-positive receptor breast cancer cell line, MCF-7. Additional analyses should be performed to evaluate whether necroptosis-related genes and their expression can play a role in modulating this effect. Additionally, more investigation into the necroptosis inflammatory process will help to understand the security of using β-glucans in cancer treatments and avoid side effects.

## Figures and Tables

**Figure 1 molecules-29-04781-f001:**
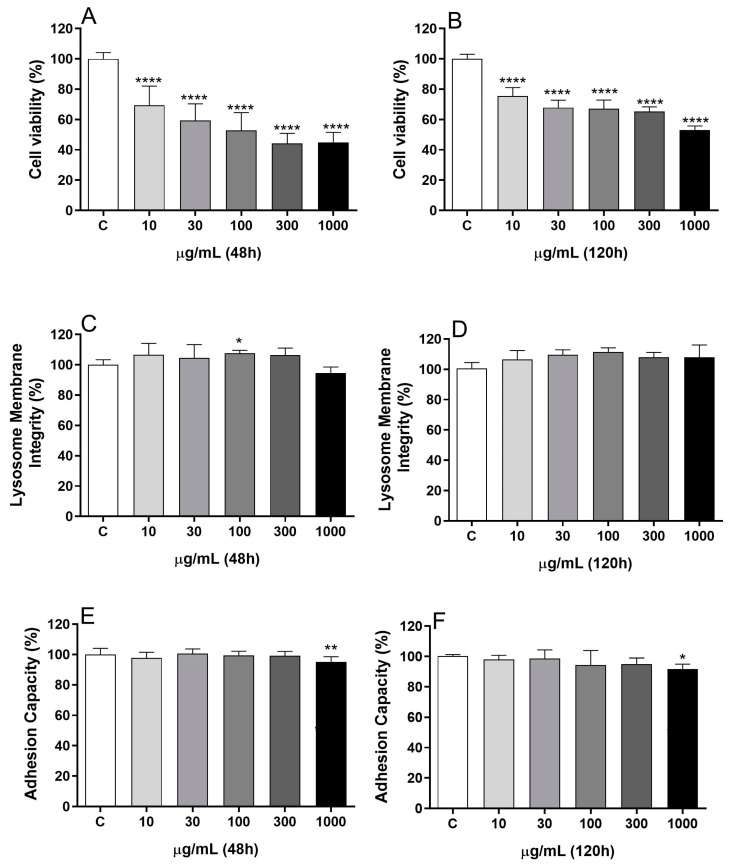
Effect of BDG16 on MCF-7 cell viability. MTT (**A**,**B**), Neutral Red (**C**,**D**), and Crystal Violet (**E**,**F**) assays were performed on MCF-7 cells after incubation with BDG16 for 48 h and 120 h. Cells were incubated with BDG16 (at 10, 30, 100, 300, and 1000 μg/mL). For the longer treatment (120 h), the cell medium was replaced by a renewed medium containing the same concentrations of BDG16 at 72 h. Statistical analyses were performed by one-way analysis of variance (ANOVA) followed by Bonferroni’s post-test, selected pairs. The results represent the mean ± SD of three independent experiments (n = 4). * *p* < 0.05; ** *p* < 0.01; **** *p* < 0.0001 compared to vehicle group (control group).

**Figure 2 molecules-29-04781-f002:**
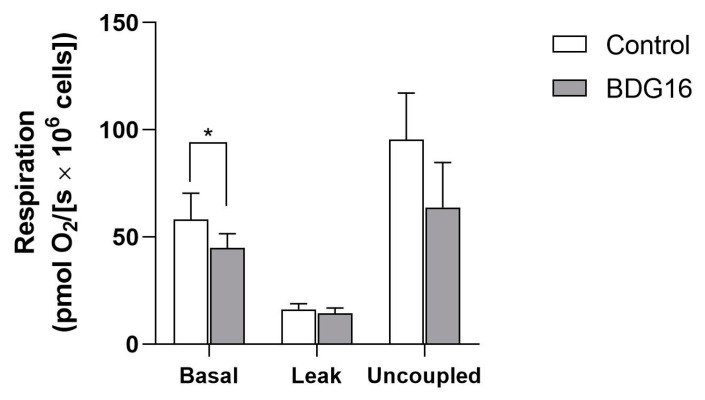
Effects of BDG16 on MCF-7 cell respiration. MCF-7 cells (1 × 10^6^ cells) were incubated with BDG16 (200 μg/mL) for 48 h. Viable cells (1.5 × 10^6^/mL) were transferred to the chambers of high-resolution oxygraph-2k, and the oxygen flux was determined in the absence of inhibitors or uncouplers (basal state), in the presence of oligomycin (leak state) and in the presence of FCCP (uncoupled state). Control: cells cultured in the absence of BDG16. The results represent the mean ± SD of four independent experiments (n = 6). * *p* < 0.05 compared to the respective control group, paired *t*-test.

**Figure 3 molecules-29-04781-f003:**
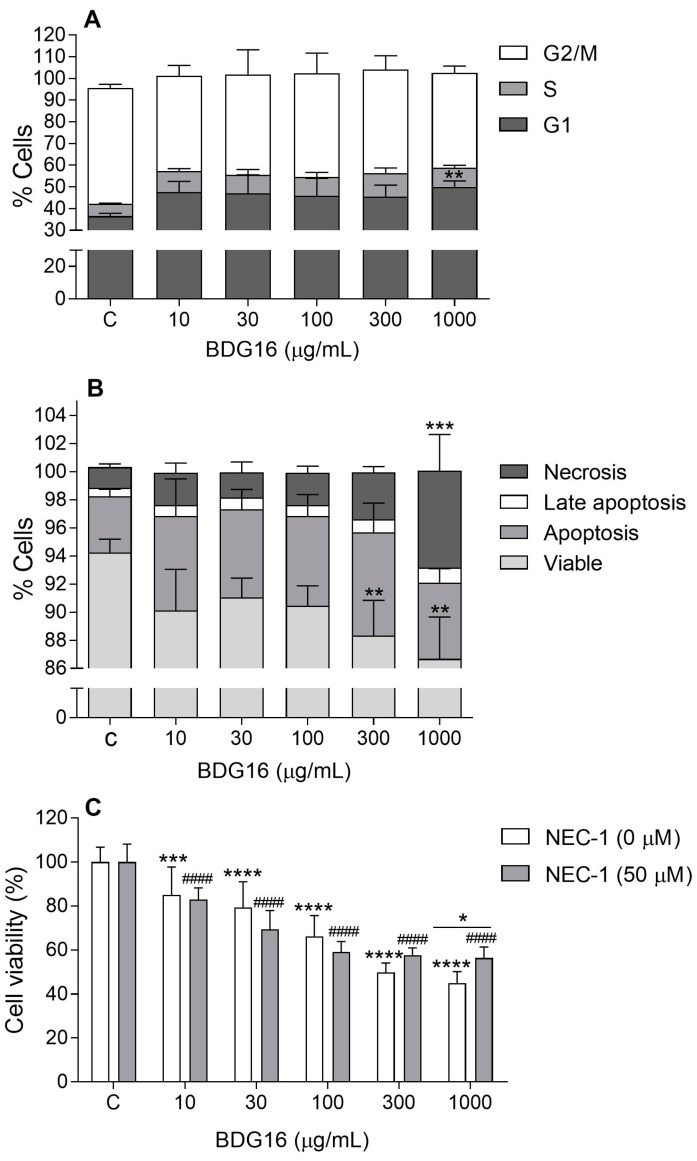
Effect of BDG16 treatment on cell-cycle progression (**A**) apoptosis/necrosis (**B**) and necroptosis (**C**). The cells were treated with BDG16 at 10, 30, 100, 300, and 1000 μg/mL for 48 h. For the necroptosis assay, MCF-7 cells were incubated with BDG16 for 48 h with or without Necrostatin-1 (NEC-1, 50 µM) previous cell viability evaluation by MTT assay. Statistical analyses were performed by one-way analysis of variance (ANOVA) followed by Bonferroni’s post-test, selected pairs. The results represent the mean ± SD of three independent experiments (n = 3). * *p* < 0.05; ** *p* < 0.01; *** *p* < 0.001; **** *p* < 0.0001 compared to vehicle group without NEC-1 (control group). #### *p* < 0.0001 compared to the vehicle group with NEC-1.

**Figure 4 molecules-29-04781-f004:**
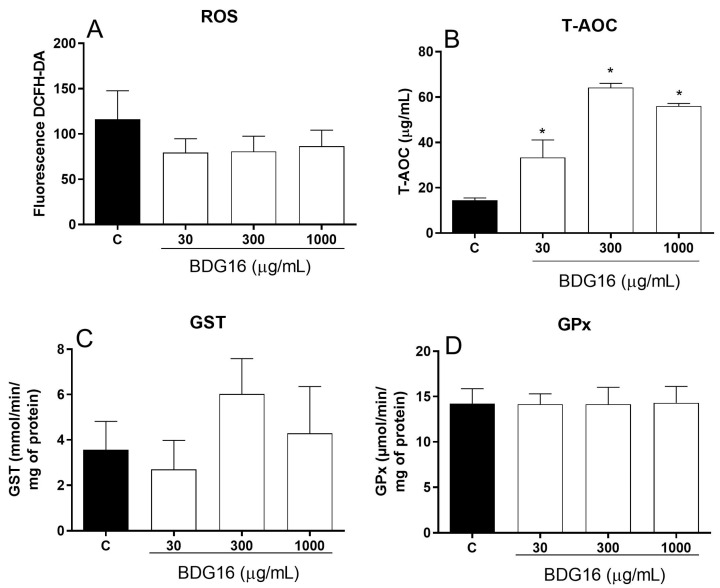
Effect of BDG16 treatment on oxidative stress. (**A**) reactive oxygen species (ROS), (**B**) Total antioxidant capacity (T-AOC), (**C**) Glutathione S-transferase (GST), and (**D**) glutathione peroxidase (GPx). The MCF-7 cells were treated with BDG16 at 30, 300, and 1000 μg/mL for 48 h. Statistical analyses were performed by one-way analysis of variance (ANOVA) followed by Bonferroni’s post-test on selected pairs. The results represent the mean ± SD of two independent experiments (n = 6), * *p* < 0.05, compared to the vehicle group (control group).

## Data Availability

Data are contained within the article.

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
