# Peer review of "Anti-Cancer Potential of Linear β-(1→6)-D-Glucan from Agaricus bisporus on Estrogen Receptor-Positive (ER+) Breast Cancer Cells"

_molecules, 2024, doi:10.3390/molecules29194781_

Round 1

Reviewer 1 Report

Comments and Suggestions for Authors

The present study revealed that β-(1→6)-D-glucan (B16) is cytotoxic to MCF-7 breast cancer cells, and its mechanism of action may be related to the inhibition of mitochondrial respiration and the induction of necrotic apoptosis. β-(1→6)-D-glucan (B16) was able to significantly reduce the metabolic activity, inhibit the cell proliferation and cell adhesion ability, and lead to the cell death of MCF-7 cells. In addition, B16 has antioxidant activity, which may be related to the cytotoxic effect of B16. The results of this study provide a new theoretical basis for the application of β-glucan in breast cancer treatment, and provide an important reference value for further research on the anti-tumor mechanism of β-glucan and the development of new therapeutic methods for breast cancer.

1. Preface mentions that B16 is a linear β-(1→6)-D-glucan, but it lacks detailed information on its primary structure, molecular weight, branching frequency, and solubility. These structural characteristics are crucial for understanding the biological activity and mechanism of action of B16. 

2. Insufficient experimental content: For instance, the description of the preparation method for β-glucan is too concise. It is recommended to supplement the detailed steps for the extraction and purification of β-(1→6)-D-glucan.

3. In the section of 3.12.4. Total Antioxidant Capacity (T-AOC) determination, it is mentioned that a 300 mM acetic acid buffer is used, but the pH value of the buffer is not specified. It is advisable to provide this information.

4. The title of this study is "Anti-tumor potential of linear β-(1→6)-D-glucan from Agaricus bisporus on breast cancer cell line," but the text only investigates the effect of linear β-(1→6)-D-glucan on estrogen receptor-positive (ER+) breast cancer cells. It is suggested to revise the title to make the research more targeted.

5. In the abstract, the research purpose of this study is not clearly stated. In the conclusion, it is recommended to prospect the clinical application of B16 in breast cancer treatment, which provides important reference value for further studying the anti-tumor mechanism of β-glucan and developing new treatments for breast cancer. 

6. Formatting issues: There are irregularities in the writing and punctuation throughout the text, and the term "P value" should be in italics. It is suggested to make the necessary corrections.

7. Problems with charts and graphs: The charts and graphs covered in the article have formatting and typographical irregularities, such as Figures 3A and B. It is suggested that they be represented in a different form to make them easier for the reader to understand.

8. Reference format: Some references are not formatted correctly. It is recommended to review and revise them.

Author Response

Dear Reviewer,

We would like to thank for your comments that helped us to improve the quality of this manuscript. All the corrections are highlighted in red in the text and responses to each topic are described below:

Comments and Suggestions for Authors

The present study revealed that β-(1→6)-D-glucan (B16) is cytotoxic to MCF-7 breast cancer cells, and its mechanism of action may be related to the inhibition of mitochondrial respiration and the induction of necrotic apoptosis. β-(1→6)-D-glucan (B16) was able to significantly reduce the metabolic activity, inhibit the cell proliferation and cell adhesion ability, and lead to the cell death of MCF-7 cells. In addition, B16 has antioxidant activity, which may be related to the cytotoxic effect of B16. The results of this study provide a new theoretical basis for the application of β-glucan in breast cancer treatment, and provide an important reference value for further research on the anti-tumor mechanism of β-glucan and the development of new therapeutic methods for breast cancer.

  1. Preface mentions that B16 is a linear β-(1→6)-D-glucan, but it lacks detailed information on its primary structure, molecular weight, branching frequency, and solubility. These structural characteristics are crucial for understanding the biological activity and mechanism of action of B16.

Response: Yes, I agree. This information was added to the first paragraph o Section 2: Results and Discussion.

  1. Insufficient experimental content: For instance, the description of the preparation method for β-glucan is too concise. It is recommended to supplement the detailed steps for the extraction and purification of β-(1→6)-D-glucan.

Response: Ok, the detailed information of how β-glucan was extracted and purified was added to the section 3.2 of the Material and Methods.

  1. In the section of 3.12.4. Total Antioxidant Capacity (T-AOC) determination, it is mentioned that a 300 mM acetic acid buffer is used, but the pH value of the buffer is not specified. It is advisable to provide this information.

Response: Yes, I agree, this information was missing. The pH value is 3.6 and it was added to the text.

  1. The title of this study is "Anti-tumor potential of linear β-(1→6)-D-glucan from Agaricus bisporus on breast cancer cell line," but the text only investigates the effect of linear β-(1→6)-D-glucan on estrogen receptor-positive (ER+) breast cancer cells. It is suggested to revise the title to make the research more targeted.

Response: Indeed, I agree with your observation. The title was changed accordingly.

  1. In the abstract, the research purpose of this study is not clearly stated. In the conclusion, it is recommended to prospect the clinical application of B16 in breast cancer treatment, which provides important reference value for further studying the anti-tumor mechanism of β-glucan and developing new treatments for breast cancer.

Response: Thank you for your suggestions, the abstract was modified to make the aim of this study clearer and more information about the clinical application of beta-glucans was provided in the last paragraph of Results and Discussion section and also in the Conclusions section.

  1. 6. Formatting issues: There are irregularities in the writing and punctuation throughout the text, and the term "P value" should be in italics. It is suggested to make the necessary corrections.

Response: Thank you for this observation, the text was carefully revised to correct these irregularities and also “P value” was corrected to be in italics.

  1. Problems with charts and graphs: The charts and graphs covered in the article have formatting and typographical irregularities, such as Figures 3A and B. It is suggested that they be represented in a different form to make them easier for the reader to understand.

Response: All the figures were revised, and the irregularities were corrected. Graphs of Fig3A and 3B were formatted in different way to make them easier to understand.

  1. Reference format: Some references are not formatted correctly. It is recommended to review and revise them.

Response: The references were revised and corrected.

Reviewer 2 Report

Comments and Suggestions for Authors

Manuscript ID: molecules-3233253

Title: Anti-tumor potential of linear β-(1→6)-D-glucan from Agaricus bisporus on breast cancer cell line

Authors: Renata Rutckeviski, Cláudia Rita Corso, Aline Simoneti Fonseca, Mariane Londero Rodrigues, Yony Román-Ochoa, Thales Ricardo Cipriani, Luciane Regina Cavalli, Silvia Maria Suter Correia Cadena, Fhernanda Ribeiro Smiderle *

The manuscript entitled „Anti-tumor potential of linear β-(1→6)-D-glucan from Agaricus bisporus on breast cancer cell lineË® describes the evaluation of in vitro anticancer potential of β-(1→6)-D-glucan on MCF-7 cells. The results could be interesting to the readers of the Molecules. However, the article can be considered for publication after a minor revision.

1.      Is it possible to change the abbreviation B16 for β-(1→6)-D-glucan ? Because B16 is an abbreviation for murine melanoma tumor cell line and in this work dealing with cell lines we have breast cancer cell line (MCF-7), so it is my suggestion to change this abbreviation, maybe, to BDG16.

2.      Page 6, line 171, Part: 2.5. Oxidative stress analyses, change ´analyses´ in ´analysis´.

3.      The suggestion is to change or modify the title. Change Anti-tumor to Anti-cancer as antitumor is more appropriate to research performed in vivo.

4.      Check the title of y-axis of Figure 4 C and 4D.

5.      Are there any results of the cytotoxicity of β-(1→6)-D-glucan on a normal cell line? Give a reference.

6.      The potential of β-(1→6)-D-glucan from Agaricus bisporus to induce necroptosis in MCF-7 cells and not apoptosis is not a preferred effect, as necroptosis induces inflammation. So this work is important in determining and reveling the mechanism of action of β-(1→6)-D-glucan, but the result of inducing necroptosis or necrosis is undesirable for further use of glucan.

Author Response

Dear Reviewer,

We would like to thank for your comments that helped us to improve the quality of this manuscript. All the corrections are highlighted in red in the text and responses to each topic are described below:

Comments and Suggestions for Authors

Manuscript ID: molecules-3233253

Title: Anti-tumor potential of linear β-(1→6)-D-glucan from Agaricus bisporus on breast cancer cell line

Authors: Renata Rutckeviski, Cláudia Rita Corso, Aline Simoneti Fonseca, Mariane Londero Rodrigues, Yony Román-Ochoa, Thales Ricardo Cipriani, Luciane Regina Cavalli, Silvia Maria Suter Correia Cadena, Fhernanda Ribeiro Smiderle *

The manuscript entitled „Anti-tumor potential of linear β-(1→6)-D-glucan from Agaricus bisporus on breast cancer cell lineË® describes the evaluation of in vitro anticancer potential of β-(1→6)-D-glucan on MCF-7 cells. The results could be interesting to the readers of the Molecules. However, the article can be considered for publication after a minor revision.

  1. Is it possible to change the abbreviation B16 for β-(1→6)-D-glucan ? Because B16 is an abbreviation for murine melanoma tumor cell line and in this work dealing with cell lines we have breast cancer cell line (MCF-7), so it is my suggestion to change this abbreviation, maybe, to BDG16.

Response: Yes, I agree. Indeed, B16 is not appropriate. The abbreviation was changed to BDG16, as suggested.

  1. Page 6, line 171, Part: 2.5. Oxidative stress analyses, change ´analyses´ in ´analysis´.

Response: Ok, the correction was made.

  1. The suggestion is to change or modify the title. Change Anti-tumor to Anti-cancer as antitumor is more appropriate to research performed in vivo.

Response: Yes, I agree. The correction was made as suggested.

  1. Check the title of y-axis of Figure 4 C and 4D.

Response: The title of y-axis was corrected, thank you for the observation.

  1. Are there any results of the cytotoxicity of β-(1→6)-D-glucan on a normal cell line? Give a reference.

Response: Yes, the results of cytotoxicity of β-(1→6)-D-glucan on a normal cell line were performed in previous study of our group, using the non-tumorigenic mammary epithelial cells MCF-10A, and are published in Rutckeviski et al., 2022 (doi: 10.1016/j.carbpol.2021.118917). This information was clarified in the text on the Item 2.1 of the results section.

  1. The potential of β-(1→6)-D-glucan from Agaricus bisporus to induce necroptosis in MCF-7 cells and not apoptosis is not a preferred effect, as necroptosis induces inflammation. So this work is important in determining and reveling the mechanism of action of β-(1→6)-D-glucan, but the result of inducing necroptosis or necrosis is undesirable for further use of glucan.

Response: Thank you for pointing out this issue. Indeed, inflammation could be a consequence of necrosis/necroptosis and this would be undesirable for further treatments with glucans. However, more investigation should be performed to clearly demonstrate these effects, especially in the whole organism (e.g. animal models), because the beta-glucans have a known immune-modulatory effect, and sometimes anti-inflammatory properties (Smiderle et al., 2008; Smiderle et al., 2014). Up to now, the few existent clinical trials in USA using oral administered beta-glucans did not relate side effects, declaring them as safe compounds. Our group is still investigating the mechanism of action, and more information was added to the discussion and conclusion pointing this out.

Smiderle, F. R., Baggio, C. H., Borato, D. G., Santana-Filho, A. P., Sassaki, G. L., Iacomini, M., & Van Griensven, L. J. L. D. (2014). Anti-Inflammatory Properties of the Medicinal Mushroom Cordyceps militaris Might Be Related to Its Linear (1→3)-β-D-Glucan. PLoS ONE, 9(10), e110266. https://doi.org/10.1371/journal.pone.0110266

Smiderle, F. R., Olsen, L. M., Carbonero, E. R., Baggio, C. H., Freitas, C. S., Marcon, R., Santos, A. R. S., Gorin, P. A. J., & Iacomini, M. (2008). Anti-inflammatory and analgesic properties in a rodent model of a (1→3),(1→6)-linked β-glucan isolated from Pleurotus pulmonarius. European Journal of Pharmacology, 597(1–3), 86–91. https://doi.org/10.1016/j.ejphar.2008.08.028